# Insights into the aetiology of snoring from observational and genetic investigations in the UK Biobank

Adrián I. Campos [1,2,6], Luis M. García-Marín[1,3,6], Enda M. Byrne[4], Nicholas G. Martin [1], Gabriel Cuéllar-Partida[2,5,7]* & Miguel E. Rentería [1,2,7]*

Although snoring is common in the general population, its aetiology has been largely understudied. Here we report a genetic study on snoring ($n \sim 408,000$; snorers $\sim 152,000$) using data from the UK Biobank. We identify 42 genome-wide significant loci, with an SNP-based heritability estimate of $\sim 10\%$ on the liability scale. Genetic correlations with body mass index, alcohol intake, smoking, schizophrenia, anorexia nervosa and neuroticism are observed. Gene-based associations identify 173 genes, including *DLEU7*, *MSRB3* and *POC5*, highlighting genes expressed in the brain, cerebellum, lungs, blood and oesophagus. We use polygenic scores (PGS) to predict recent snoring and probable obstructive sleep apnoea (OSA) in an independent Australian sample ($n \sim 8000$). Mendelian randomization analyses suggest a potential causal relationship between high BMI and snoring. Altogether, our results uncover insights into the aetiology of snoring as a complex sleep-related trait and its role in health and disease beyond it being a cardinal symptom of OSA.

[1] Genetic Epidemiology Lab, QIMR Berghofer Medical Research Institute, Brisbane, QLD, Australia. [2] Faculty of Medicine, The University of Queensland, Brisbane, QLD, Australia. [3] Tecnológico de Monterrey, Escuela de Ingeniería y Ciencias, Zapopan, Jalisco, México. [4] Institute for Molecular Bioscience, The University of Queensland, Brisbane, QLD 4072, Australia. [5] University of Queensland Diamantina Institute, Brisbane, QLD, Australia. [6] These authors contributed equally: Adrián I. Campos, Luis M. García-Marín. [7] These authors jointly supervised this work: Gabriel Cuéllar-Partida, Miguel E. Rentería. *email: g.cuellarpartida@uq.edu.au; miguel.renteria@qimrberghofer.edu.au

Snoring is the vibration of the upper airway structures that occurs during sleep and creates noise as the air passes in and out while breathing. Habitual snoring is common in the population, its overall prevalence increases with age and is higher in males (35–45%) than females (15–28%)[1]. Importantly, snoring is a hallmark of obstructive sleep apnoea (OSA), a sleep-related breathing disorder characterized by repeated episodes of complete or partial obstructions of the upper airway during sleep, despite the effort to breathe[1]. OSA is usually associated with a reduction in blood oxygen saturation and is often accompanied by associated daytime symptoms, such as excessive daytime sleepiness, fatigue and decreased cognitive function. Although the vast majority of patients with OSA exhibit snoring, a minority (20–25%) of patients with central sleep apnoea do not snore[2] and it is estimated that sleep apnoea may occur in as many as 20–40% of the adult population that are snorers, leaving the remaining 60–80% of snorers in the category of habitual non-apnoeic benign snorers. Snoring has previously been associated with body mass index (BMI)[3,4] as well as with the risk of cardiovascular disease such as coronary heart disease and stroke among postmenopausal women[5]. Twin and family studies have demonstrated the existence of a genetic predisposition to habitual snoring, with heritability estimates suggesting that 18–28% of variance can be accounted for by genetic factors[4,6]. A proportion of its heritability may be mediated through other heritable lifestyle factors such as smoking and alcohol consumption, which can also contribute to snoring[7–9].

Snoring is known to reduce sleep quality for both snorers and their sleeping partners[10,11], reducing energy/vitality and increasing daytime anxiety[12], risk of depression, stress, fatigue and sleepiness[11]. Here we leverage data from the UK Biobank and an Australian sample of adults, in an effort to characterize the molecular underpinnings of habitual snoring as a complex, polygenic trait. We estimate the prevalence of snoring at 36% and identify phenotypic correlations with BMI, socio-economic status (SES), smoking and alcohol consumption frequency. A genome-wide association study (GWAS) identifies 42 genome-wide significant loci and a significant single-nucleotide polymorphism (SNP)-based heritability of ~10%. We perform sensitivity analyses, which suggest that the genetic aetiology of snoring is not uniquely driven by BMI genetic factors. We further assess the existence of sex-specific effects, identifying two loci with evidence of differential effect sizes, albeit in the same direction, between males and females. No large-scale sex-specific genetic effects are identified. We also employ statistical genetics methods such as linkage disequilibrium (LD) score regression and Mendelian randomization (MR), to identify genetically correlated traits with snoring and assess causality. Sleep-related traits such as sleep apnoea, BMI, cardiometabolic and psychiatric traits are genetically correlated with snoring. MR analyses suggest a potential causal relationship between higher BMI and snoring. Furthermore, we use our GWAS results to estimate individual polygenic scores (PGS) and predict snoring and probable OSA in an independent sample of Australian adults, highlighting the possibility of studying this complex sleep disorder using snoring as a proxy phenotype.

## Results

**Snoring prevalence and risk factors**. Our population-based discovery sample consisted of 408,317 individuals of white British ancestry from the UK Biobank. Participants in the sample were deemed as snoring 'cases' (37%) based on their report that a partner or housemate had complained to the participant about their snoring (see Methods and Table 1). Snoring was significantly associated with age (odds ratio (OR) = 1.011 [per year,

95% confidence interval (CI) 1.009–1.012]) and, to a greater extent, with sex ($OR_{males}$ = 2.264 [2.212–2.316]). The prevalence of sleep apnoea was higher within the snorer group (Table 1). Furthermore, BMI, SES, smoking frequency and alcohol consumption frequency were also associated with snoring (Fig. 1a). Although snoring prevalence was higher in males, BMI was positively correlated with snoring prevalence in both males and females (Fig. 1b). A lower SES, as determined by both Townsend deprivation index and average household income, was associated with increased snoring in males only. Smoking frequency was positively correlated with snoring prevalence in females and, to a lesser extent, in males (Fig. 1a–c). In contrast, alcohol consumption frequency was correlated with snoring in males and, to a lesser extent, in females (Fig. 1a–d). We further identified other factors such as whole-body fat mass and sleep duration that are correlated with snoring (Supplementary Table 1).

**Discovery GWAS and SNP heritability**. We performed a GWAS study of snoring, taken as a dichotomous variable ($n = 408,317$; cases ~152,000; controls ~256,000). After quality control (QC; see Methods), 11,010,159 genetic variants remained in the analysis. This uncovered 127 independent genome-wide significant associations across 41 genomic risk loci (Fig. 2a and Supplementary Fig. 1)[13]. Annotation for the top 15 risk loci is shown in Table 2 and a list of all genomic risk loci is given in Supplementary Data 1. The overall SNP heritability on the liability scale ($h^2_{SNP}$) was 9.9% (SE = 0.39%).

**Genetic correlations**. The trait that showed the highest genetic correlation with habitual snoring was self-reported sleep apnoea ($r_G = 0.78$, SE = 0.17, $p$-value = $3 \times 10^{-05}$ [$ldsc$ $\chi^2$-test]; Supplementary Data 2). We also analysed the genetic correlation between snoring and three measures of overnight oxyhemoglobin saturation: average $SpO_2$, minimum $SpO_2$ saturation and percent of sleep with oxyhemoglobin saturation under 90% (Perc90)[14]. Minimum $SpO_2$ and Perc90, which are known proxies for sleep-disordered breathing, but not average $SpO_2$ (which reflects changes in ventilation not necessarily related to sleep apnoea), showed moderate significant genetic correlations with snoring (Fig. 3). Other traits genetically correlated to snoring included BMI, whole-body fat mass, sodium in urine, mood swings, coronary artery disease, alcohol intake frequency, pulse rate, current tobacco smoking, heart disease, lung cancer, the ratio between forced expiratory volume in 1 s (FEV1) and forced vital capacity (FVC), neuroticism, subjective wellbeing and heart rate, among others. Traits showing a negative genetic correlation with snoring included schizophrenia, FVC, FEV1, fluid intelligence score, educational attainment, age at menarche, mean accumbens volume and anorexia nervosa. Overall, traits related to BMI, risk for psychiatric disease, lung function and heart disease were among those with the strongest evidence of association (Fig. 3 and Supplementary Data 2). Notably, pulse rate, whole-body fat mass and BMI were also phenotypically associated with snoring in this sample (see above and Supplementary Table 1).

**Sensitivity analysis**. We performed two follow-up sensitivity GWAS to explore the effects of BMI, adiposity and nonlinear effects on associated variants. The first sensitivity GWAS included BMI as a covariate, whereas the second included BMI, $BMI^2$, age × sex, $age^2$ and whole-body fat mass. Both sensitivity analyses showed very similar results, with a genetic correlation of 0.9998 (SE = 0.0002). We therefore focus below on the simple model adjusting only for BMI and basic covariates (see Methods). The results revealed 97 genome-wide significant SNPs across 34 genomic risk loci (Fig. 2a) with overall SNP heritability on the

**Table 1 Sample composition and descriptive statistics of UK Biobank discovery sample.**

|  | Female N (%) | Apnoea N (%) | Age mean (SD) | BMI mean (SD) |
|---|---|---|---|---|
| Cases (snorers) | 63833 (40.74%) | 4510 (2.88%) | 57.01 (7.70) | 28.67 (4.85) |
| Controls | 161775 (61.44%) | 1663 (0.63%) | 56.60 (8.21) | 26.64 (4.52) |
| Total | 225608 (53.72%) | 6173 (1.47%) | 56.75 (8.03) | 27.39 (4.75) |

N = Sample sizes. Descriptive statistics were calculated only for the subset of the data with European or British ancestry.

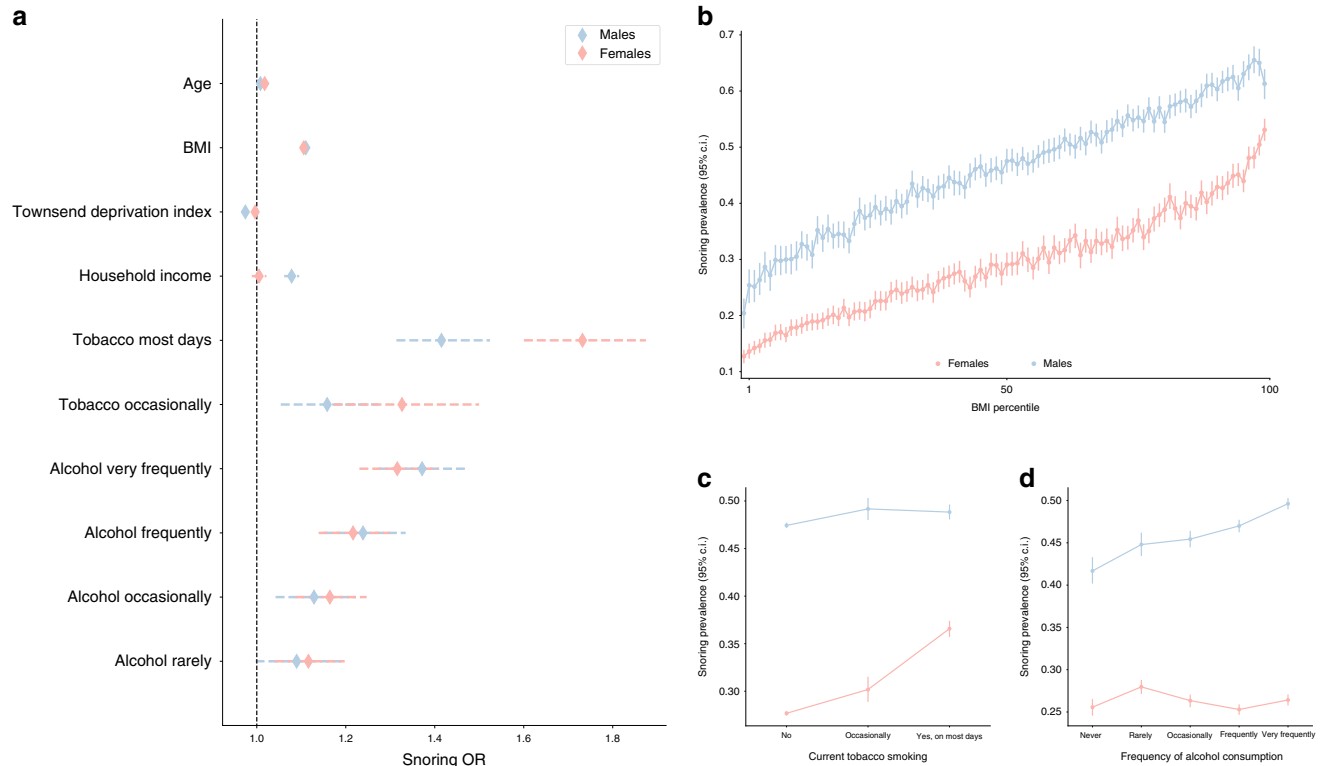

**Fig. 1 Lifestyle and demographic factors associated with increased snoring. a** Forest plot depicting the odds ratios of studied variables on snoring for males (blue) and females (red). **b** Sex- and BMI-stratified prevalence of snoring. **c** Sex- and frequency of tobacco smoking-stratified snoring prevalence. **d** Sex- and frequency of alcohol consumption-stratified snoring prevalence. Error bars denote the 95% confidence intervals estimated using the odds ratio and SE (**a**) or 1000 pseudo replicates from a bootstrap resampling procedure (**b**, **c**, **d**).

liability scale ($h^2_{SNP}$) of 8.67% (SE = 0.39) (Table 3). Although most of the lead SNP effect sizes remained very similar, the signal in the *FTO* locus showed a strong change after BMI adjustment (Fig. 2b and Supplementary Data 1). Further, after adjusting for BMI, snoring was no longer genetically correlated with BMI and adiposity-related phenotypes, nor with heart attack, hypertension, alcohol and smoking habits. Traits that remained genetically correlated with snoring after adjusting for BMI included schizophrenia, educational attainment, sodium in urine and sleep-related traits such as daytime dozing, sleep apnoea and excessive daytime sleepiness. Notably, measures of nocturnal hypoxia such as minimum $SpO_2$, average $SpO_2$ and Perc90 showed small increases in their genetic correlation with snoring after adjusting for BMI (Fig. 3 and Supplementary Data 3). The genetic correlation between both the adjusted and unadjusted GWAS was high ($r_G = 0.923$, SE = 0.003, $p$-value = $1 \times 10^{-300}$), suggesting that a considerable amount of snoring predisposition is not fully explained by BMI.

**Positional, eQTL and gene-based test prioritization**. To gain insights into the functional consequences of individual

genome-wide significant variants, we used positional and expression quantitative trait loci (eQTL) mapping, as well as genome-wide gene-based association analyses. From positional and eQTL mapping, we identified 149 protein-coding genes mapping to a genome-wide significant SNP. The nearest genes to the top signals included *DLEU7* on chromosome 13 and *MSRB3* on chromosome 12. In addition to *DLEU7* and *MSRB3*, other compelling genes (prioritized by positional or eQTL mapping) for snoring included *BCL11B*, *FTO*, *SMG6*, *ROBO2*, *NSUN3*, *SNAP91* and *BCL2*, which have previously been associated with smoking[15,16]; *BLC11B*, *FTO*[17], *RNA5SP471*[17,18] and *SND1* and *NSUN3*, previously associated with alcohol consumption[15,17–19]; *FTO* and *SND1*, associated with coffee consumption[20]; *LMO4* associated with insomnia[21]; and *RNA5SP471* with narcolepsy[21,22]. In addition, *ROBO2* was previously associated with chronotype[23,24] and multiple genes (*DLEU7*, *MSRB3*, *FTO*, *ANAPC4*, *SMG6*, *SND1*, *SIM1*, *KCNQ5*, *CEP120*, *MACF1*, *SNAP91* and *BCL2*) previously associated with musculoskeletal traits such as height and heel bone mineral density (Supplementary Data 1 and 2)[25–28]. Genome-wide gene-based association analysis identified 179 genes associated with snoring beyond genome-wide significance

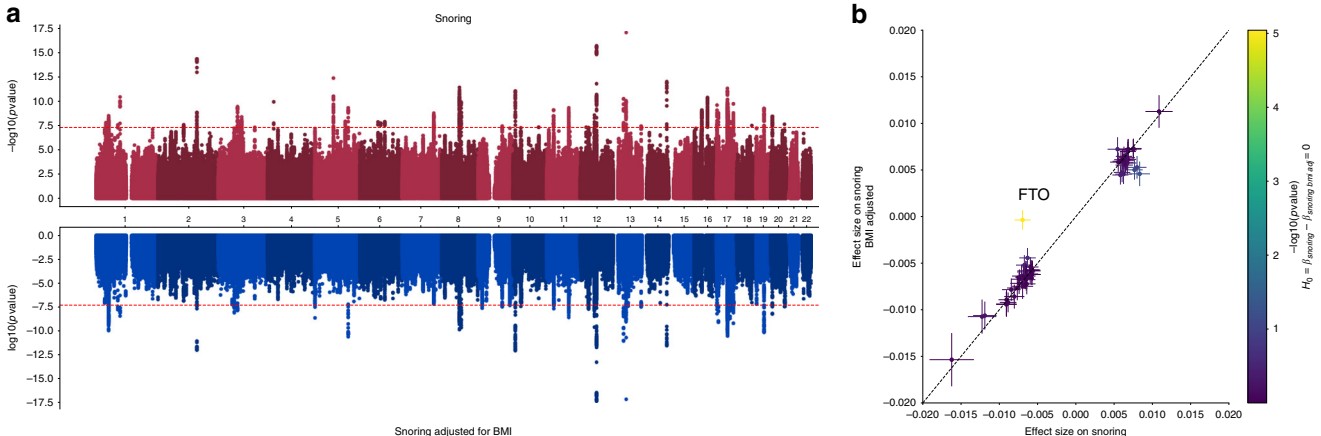

**Fig. 2 Genetic variants associated with snoring with and without adjustment for BMI. a** Results from the genome-wide association studies are presented as a Miami (mirrored Manhattan) plot. The x-axis contains a point for each genetic variant passing QC and is ordered by chromosome and base position. The distance between each variant and the x-axis is a function of the significance (p-value) of the association. For the top panel the $-\log10(p\text{-value})$ is graphed on the y-axis, whereas the $\log10(p\text{-value})$ is shown for the bottom panel (BOLT-LMM mixed model association test p-value). The red dotted line denotes the genome-wide significance threshold ($p < 5e - 8$). **b** Effect sizes of independent lead SNPs across snoring (x-axis) and snoring adjusted for BMI (y-axis). The only locus (FTO) with a statistically significant difference is labelled.

**Table 2 Top 15 genomic risk loci for snoring showing the top SNP for each locus.**

| SNP | Chr | Position | NEA | EA | MAF | Nearest gene | gwasP | β | SE |
|---|---|---|---|---|---|---|---|---|---|
| rs592333 | 13 | 51,340,315 | G | A | 0.4423 | *DLEU7* | $1.00E - 17$ | −0.00906 | 0.001051 |
| rs10878269 | 12 | 65,791,463 | T | C | 0.3499 | *MSRB3* | $2.30E - 16$ | −0.00886 | 0.001086 |
| rs61597598 | 2 | 156,996,626 | A | G | 0.1163 | *AC073551.1* | $5.10E - 15$ | −0.01189 | 0.001529 |
| rs2307111 | 5 | 75,003,678 | C | T | 0.3956 | *POC5* | $4.80E - 13$ | 0.007667 | 0.00107 |
| rs2664299 | 14 | 99,742,187 | C | T | 0.4145 | *BCL11B* | $1.10E - 12$ | 0.007503 | 0.001061 |
| rs13251292 | 8 | 71,474,355 | G | A | 0.4145 | *TRAM1* | $4.30E - 12$ | −0.00737 | 0.001067 |
| rs57222984 | 17 | 43,758,898 | G | A | 0.2654 | *CRHR1:RP11-105N13.4* (ncRNA) | $5.40E - 12$ | −0.00843 | 0.00122 |
| rs725861 | 10 | 9,063,776 | G | A | 0.1918 | *RP11-42L9.2* | $1.00E - 11$ | −0.00908 | 0.001338 |
| rs12119849 | 1 | 96,878,072 | A | G | 0.0825 | *UBE2WP1* (pseudogene) | $4.10E - 11$ | −0.01226 | 0.00186 |
| rs796856741 | 16 | 53,799,278 | GT | G | 0.4433 | *FTO* | $4.70E - 11$ | −0.00696 | 0.001059 |
| rs12429765 | 13 | 40,745,860 | G | A | 0.493 | *LINC00332* | $6.20E - 11$ | 0.0068 | 0.001051 |
| rs34811474 | 4 | 25,408,838 | A | G | 0.2167 | *ANAPC4* | $1.30E - 10$ | 0.007996 | 0.001237 |
| rs7829639 | 8 | 78,215,352 | G | A | 0.2972 | *AC105242.1* (miRNA) | $1.40E - 10$ | −0.00741 | 0.001155 |
| rs180107 | 17 | 67,930,772 | T | A | 0.3698 | *AC002539.2* (miRNA) | $2.10E - 10$ | −0.0068 | 0.00106 |
| rs11409890 | 17 | 46,269,542 | TA | T | 0.4821 | *SKAP1:RP11-456D7* | $2.20E - 10$ | 0.006664 | 0.001061 |

($p < 2.636e - 6$; Bonferroni-corrected threshold for 18,971 tested genes) several of which were consistent with the mapped genes. After adjusting for BMI, 104 protein-coding genes were identified mapping to a genome-wide significant SNP from the positional and eQTL mapping, whereas 120 genes remained significantly associated with snoring, including both *MSRB3* and *DLEU7* (see Supplementary Fig. 2 and Supplementary Data 2 and 3). eQTL data obtained from Genotype-Tissue Expression (GTEx) and mapped with Functional Mapping and Annotation of Genome-Wide Association Studies (FUMA) highlighted significant SNPs that were associated with the expression of genes in several tissues including the lungs, blood, oesophagus, breast mammary, tibial nerve, and several areas of the brain, such as the cerebellum and hippocampus (Supplementary Fig. 3 and Supplementary Data 2). In summary, many of the mapped genes for snoring have been previously associated with other traits and diseases, primarily grouped into cardiometabolic, cognitive/neurological, respiratory and psychiatric (Fig. 4 and Supplementary Data 1).

To further assess whether significant genes converged in functional gene sets and pathways, we conducted gene-set enrichment analyses of tissue expression data (Supplementary Fig. 3a). Genes expressed in blood vessel and tibial artery tissue were associated with snoring, even after adjusting for BMI (Supplementary Fig. 3b). Given these associations, and an observed genetic correlation between snoring and pulse rate ($r_G = 0.106$, SE $= 0.03$, p-value $= 0.001$), we conducted a two-sample generalized summary-data-based MR (GSMR)[29] to test for a possible causal relationship. The analysis suggested a one-way causal relationship in which snoring increased pulse rate (Supplementary Fig. 4). We further explored the association between snoring and BMI, whole-body fat mass, blood pressure, major coronary heart disease and heart attack. GSMR results suggested a bidirectional causal relationship, with snoring exerting a causal effect on BMI, but also BMI exerting a causal effect on snoring, and a similar pattern was observed for heart attack. In addition, one-way causal relationships were seen for whole-body fat mass causing snoring and for snoring causing an increase in blood pressure (Supplementary Fig. 4). To control for possible confounding due to sample overlap, we conducted GSMR analyses using sex-stratified GWAS results (see Methods). The results supported causal relationships between BMI (and whole-body fat mass) causing snoring, whereas all

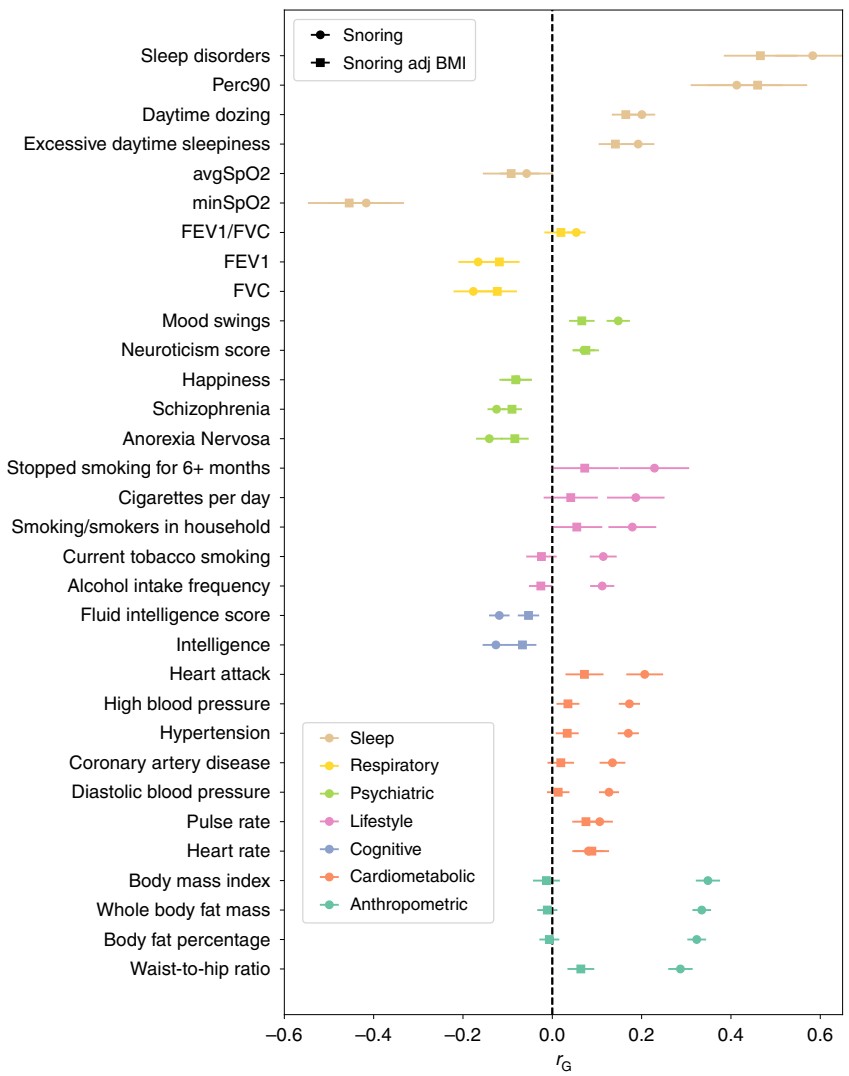

**Fig. 3 Snoring is genetically correlated with lifestyle, psychiatric and respiratory traits.** LD score-based estimates of the genetic correlation between snoring (circles) or snoring adjusted for BMI (squares) and other complex traits (y-axis). All of the depicted traits had a statistically significant genetic correlation with snoring after multiple testing correction (Benjamini–Hochberg; FDR < 0.05). Error bars show the SE of the genetic correlations. Not all tested associations are shown due to lack of space; complete results are available in Supplementary Data 2.

**Table 3 SNP-based heritability of snoring on the liability scale.**

| Trait | h2SNP | SE | λGC | Intercept |
|---|---|---|---|---|
| Snoring | 9.9% | 0.39% | 1.428 | 1.04 (0.01) |
| Snoring adj. for BMI | 8.67% | 0.39% | 1.368 | 1.03 (0.009) |
| Snoring males | 8.77% | 0.54% | 1.200 | 1.01 (0.007) |
| Snoring females | 12.42% | 0.57% | 1.254 | 1.02 (0.007) |
| Snoring adj. for BMI males | 7.72% | 0.56% | 1.200 | 1.01(0.008) |
| Snoring adj. for BMI females | 10.85% | 0.54% | 1.253 | 1.02 (0.007) |

LD score regression derived SNP-based heritability results. Estimates were transformed to the liability scale, assuming equal population and sample prevalence. $\lambda_{GC}$ is the genomic inflation factor and intercept is the LD score regression intercept.

other associations did not reach statistical significance after controlling for multiple testing (Supplementary Table 2). We performed further sensitivity analyses using five different MR methods that test different assumptions (Fig. 5). The results of GSMR, inverse variance weighted (IVW-MR) and weighted

median analyses supported a causal effect of BMI (and whole-body fat mass) on snoring. Notably, although the MR-Egger estimates did not reach statistical significance, the Egger intercept was not significantly different from zero.

**Sex-stratified GWAS.** Given the higher prevalence of snoring in males, we conducted GWAS analyses stratified by sex. These analyses identified 4 and 25 genome-wide significant SNPs for snoring in males and females, respectively. SNP heritability on the liability scale ($h^2_{SNP}$) was 8.77% (SE = 0.54%) and 12.42% (SE = 0.57%), respectively, for males and females (Table 3). In the sensitivity analyses, SNP heritability ($h^2_{SNP}$) was slightly lower after adjusting for BMI in both males 7.72% (SE = 0.56%) and females 10.85% (SE = 0.54%) (Table 3). We identified two loci (lead SNPs rs199797821 and rs200391180) with a significantly different effect size between sexes, although in the same direction. The cross-sex genetic correlation was high ($r_G = 0.914$, SE = 0.033, p-value = $1.91 \times 10^{-160}$), and effect sizes and directions for top hits were highly consistent in both the male and female samples (Supplementary Data 1 and Supplementary Fig. 5).

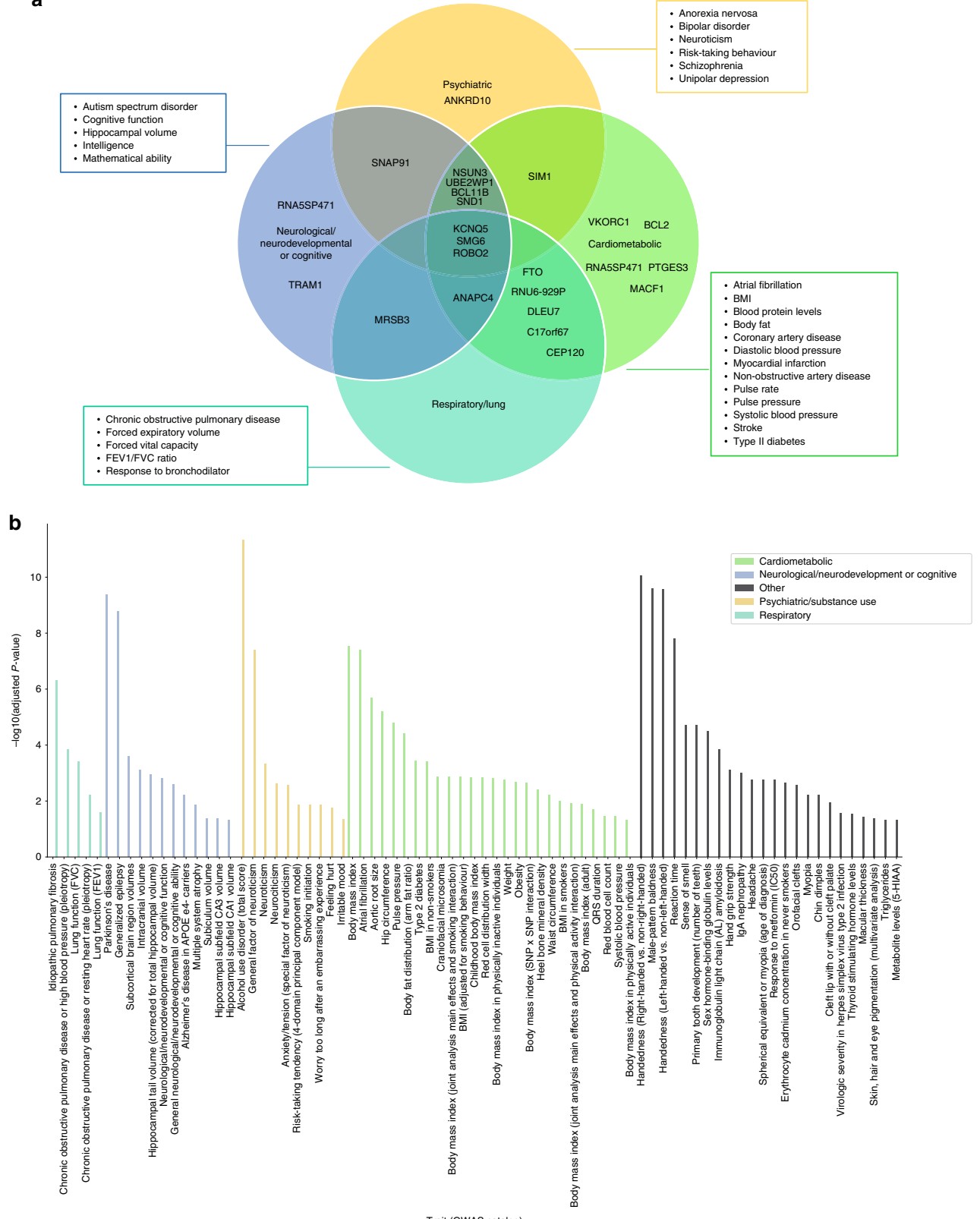

**Fig. 4 Snoring genes associated with other traits or diseases. a** Venn diagram showing the nearest genes to the lead significant SNP per genomic risk loci identified for snoring, categorized according to previously reported association with other traits or diseases in the GWAS catalogue. **b** Significant gene-set enrichment analysis (hypergeometric test) based on all prioritized genes against gene sets defined by traits in the GWAS catalogue.

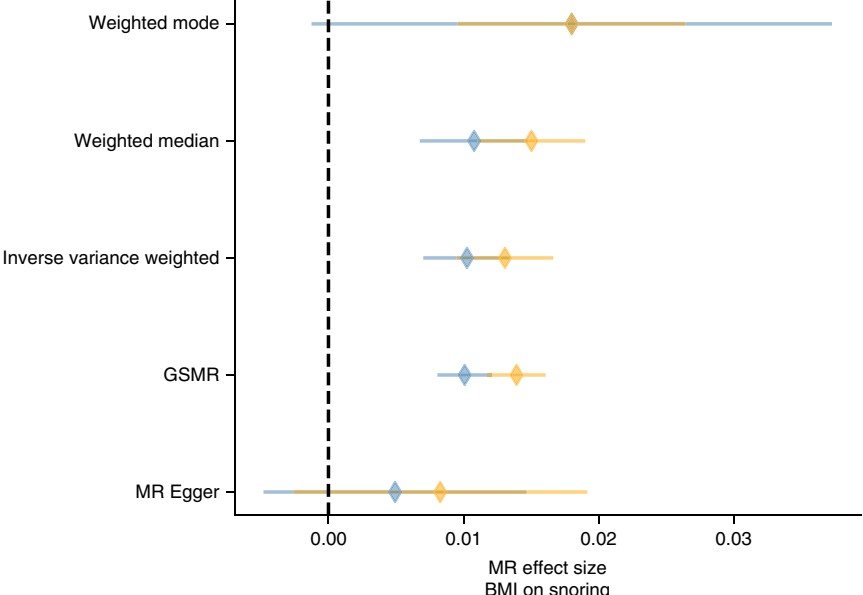

**Fig. 5 Assessing the causal relationship between BMI and snoring.** Forest plot showing Mendelian randomization results testing for causal relationships between BMI and snoring. Blue markers represent the estimate between the exposure effect sizes (female-only GWAS) and assessing its causal relationship on snoring (male-only GWAS). Yellow markers show the exposure effect sizes (male-only GWAS) assessing its causal relationship on snoring (female-only GWAS). Diamonds represent the effect size and error bars represent the 95% confidence intervals.

**PGS and prediction on an independent sample.** We used the discovery GWAS summary statistics to derive PGS in an independent target sample from the Australian Genetics of Depression Study (AGDS)[78]. The prevalence of self-reported recent snoring was 32%, with a higher prevalence in males than in females (43.2% and 28.1%, respectively). PGS for snoring were significantly associated with recent snoring for all but one ($p <= 5e-8$) of the $p$-value inclusion thresholds (Fig. 6 and Supplementary Fig. 6). Participants in the highest snoring PGS decile had around twice the odds of reporting recent snoring and choking or struggling for breath during sleep (i.e., probable OSA; sample prevalence = 8.2%) compared with those on the lowest decile (Fig. 6a). Furthermore, the PGS showed a stronger association with increasing frequency of snoring severity (Fig. 6b). Finally, we showed that the snoring PGS explained a significant amount of variance in recent snoring (Fig. 6c).

## Discussion
This study advances our understanding of the aetiology and genetic architecture of snoring. Overall, snoring prevalence was higher in males than in females, having a strong and positive correlation with BMI, tobacco smoking and alcohol consumption in both sexes. The effects of BMI, smoking and alcohol consumption have been previously reported in other studies[7,30,31]. In our study, tobacco smoking displayed a stronger association with snoring in females compared with males, a result consistent with observations in an independent sample of ~15,000 Europeans published in 2004[31]. Previous studies provided evidence that women might have a greater susceptibility to chronic obstructive pulmonary disease after smoking[32] and a greater bronchial hyperresponsiveness after methacholine challenge[33]. These results suggest that smoking might be associated with snoring through an increased inflammatory response and irritation of the airways, thus having a larger effect size on females, compared with males. Conversely, our study indicates that the frequency of alcohol consumption has a stronger influence on snoring in males compared with that in females. This is consistent with a previous study that failed to identify an association between alcohol

consumption and snoring in females but did so in males[34]. Our results strengthen the evidence pointing to alcohol as a risk factor for snoring and sleep apnoea, potentially through a weakening (relaxation) effect in the jaw and pharyngeal muscles[35]. Our study revealed that lower SES was associated with higher snoring prevalence in males only. Considering that the analysis simultaneously accounted for the effects of BMI, alcohol and tobacco consumption frequency, we speculate that the snoring–SES association might be mediated through factors that are associated with a lower SES and differ between males and females (e.g., work-related exposures). Nonetheless, whether SES is causally associated with snoring in males remains to be assessed. This would require well-powered genetic correlates of SES on independent samples. The differences in risk factor effect sizes between males and females might contribute to the overall observed sex difference in snoring prevalence. Future studies should leverage statistical genetics methods such as polygenic scoring or MR to further characterize the role of SES, smoking and alcohol-related phenotypes in snoring and OSA.

The sex differences described above motivated us to perform sex-stratified GWAS. The larger sample size of the female subgroup conferred more power to detect genetic associations in our analyses. Notably, we identified a higher snoring SNP-based heritability in females than in males and two loci that displayed statistically significant different effect sizes between sexes. Nonetheless, the observed high cross-sex genetic correlation and a high concordance in effect size and direction among top hits suggest that differences in sex-stratified GWAS might be due to power differences between the male and female subsamples rather than the existence of large-scale sex-specific genetic effects. Future studies should assess whether the loci with evidence of sex-specific effects are mediating the differential effects of SES, alcohol or tobacco consumption frequency between sexes.

Top genes identified from gene-based test analysis for snoring included *DLEU7* and *MSRB3*. Previous reports have associated *DLEU7* with heel bone mineral density[25,36], BMI[37,38], height[39,40], cardiovascular diseases[41], systolic blood pressure[41] and pulmonary function decline (FEV)[42]. The association between snoring

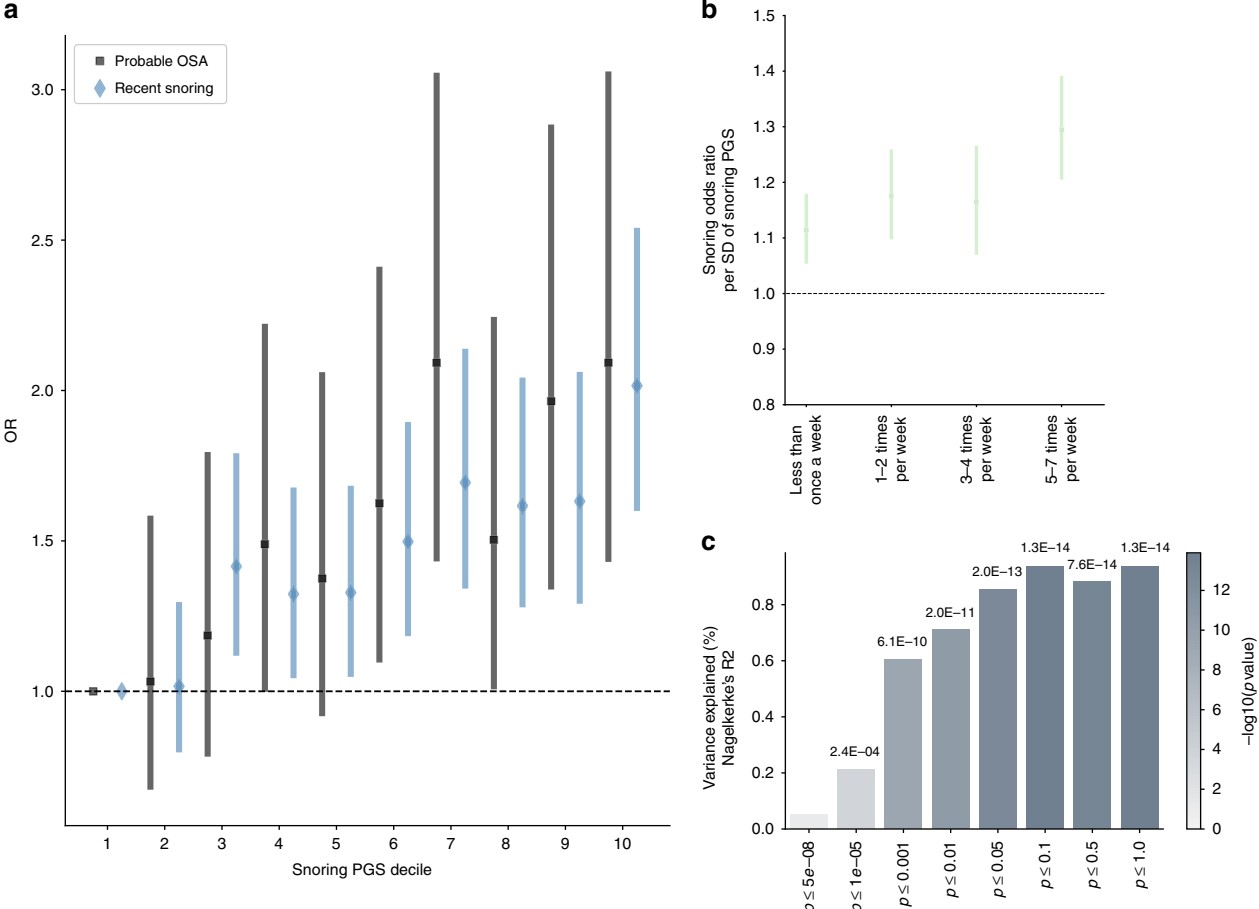

**Fig. 6 Polygenic scores predict snoring and probable apnoea in an independent sample. a** Forest plot showing the odds ratios (and 95% CI) by decile of polygenic score (PGS) for snoring from the UK Biobank discovery sample (relative to the bottom decile = 1) for recent snoring and probable sleep apnoea measured in an independent target sample of ~8000 unrelated Australian adults from the Australian Genetics of Depression Study (AGDS). **b** Snoring PGS predicting different snoring frequencies reported in AGDS target sample. **c** Variance of recent snoring in AGDS explained by PGS calculated from UK Biobank summary statistics. The x-axis represents the p-value threshold used for variant inclusion during genetic scoring; the y-axis represents the amount of variance explained (change in Nagelkerke $R^2$). The colour of each bar represents the significance of the association between the PGS and recent snoring (−log10 p-value), while the exact p-value (Wald's test) is shown above each bar.

genes and heel bone mineral density could be mediated by BMI due to the association between BMI and bone density documented previously[43]. *MSRB3* plays a relevant role in protein and lipid metabolism pathways[44], and has been associated with hippocampal volume[45–47], lung function[41,48,49], Alzheimer's disease[50,51], brain injuries[52], novelty seeking[53], deafness[54] and height[41]. These results could be consistent with the fact that severe snoring may incur in nocturnal oxygen desaturation[55], diminishing neuropsychological functions and, in some cases, resulting in tissue damage[56] and contributing to impairment of memory consolidation processes[57]. However, more research is needed to test this hypothesis.

Genetic correlations between snoring and a variety of traits and diseases were identified. Sleep-related traits, among others, survived the sensitivity analysis adjusting for BMI. It is important to point out that adjustment for a heritable covariate can not only reduce power for estimating genetic correlations or causality estimates, but also introduce collider bias[58]. We have therefore only discussed genetic correlations that existed before adjusting for BMI. The traits with the highest genetic correlation with snoring were sleep apnoea and sleep-disordered breathing phenotypes, which is consistent with loud snoring being a diagnostic criterion for OSA. This observation is also remarkable given the small sample size (<2000 cases) and therefore reduced power (no

genome-wide hits) that the GWAS for self-reported sleep apnoea has in the UK Biobank. Our analyses suggest that a GWAS for snoring captures a substantial portion of the genetic contribution to sleep apnoea, highlighting the importance of studying symptoms on a subclinical threshold, an approach that has already proven useful at understanding the heterogeneity of other complex traits such as depression and neuroticism[59,60]. Our study will enable future efforts aimed at understanding the underlying genetic architecture of OSA using multivariate statistical genetic approaches.

We also observed moderate correlations with BMI, obesity and whole-body fat mass. Other relevant correlations included lung function, neurological, cardiovascular and psychiatric diseases, and traits such as alcohol consumption frequency and smoking. This is consistent with the observed phenotypic associations on the first part of this study. The high genetic correlation between snoring and snoring adj. BMI ($r_G = 0.923$, SE = 0.003, p-value = $1 \times 10^{-300}$) supports the idea that the genetic architecture of snoring cannot be explained simply by BMI. Notably, the genetic correlations between snoring and diseases such as asthma and allergic rhinitis, which are considered risk factors for sleep-disordered breathing[61], do not reach statistical significance. This could imply that the association of atopic diseases and sleep-disordered breathing is not mediated through genetics, but future

studies should assess this in a more systematic manner. Our results highlight the importance and utility of studying snoring, and unveil opportunities for understanding highly related sleep traits and disorders, including OSA.

Our initial MR results using GSMR suggested a mutual causal relationship between BMI and snoring, and a similar pattern was observed for heart attack, but only a one-way causal relationship of whole-body fat mass causing snoring. We hypothesized BMI to be more heterogeneous and potentially more pleiotropic than whole-body fat mass. In fact, MR is known to be confounded by pleiotropy[62]. Interestingly, a one-way causal relationship between snoring and pulse rate, which survived adjustment for BMI, was identified. Nonetheless, this association did not reach statistical significance when accounting for sample overlap (i.e., sex-stratified GSMR) or when using other MR methods that also account for pleiotropy. The only causal associations that survived sample overlap and multiple testing correction were BMI or whole-body fat mass causing snoring. Evidence for a causal association was also observed from methods such as IVW-MR and weighted median MR in addition to GSMR. Using more stringent methods such as MR-Egger, the association did not retain statistical significance. Nonetheless, it is known that MR-Egger is a less powered method[63]. Furthermore, the MR-Egger intercept was not significantly different from zero, thus suggesting that the IVW-MR causality estimate is likely to be unbiased[64]. In addition, GSMR removes pleiotropic instruments using a HEIDI outlier filter and should be unbiased by pleiotropy[29]. Overall, we believe this to be a compelling evidence of a causal effect of BMI on snoring, but caution should be taken given the results from MR-Egger and when extrapolating this observation to other related traits such as OSA. The lower number of instruments available for snoring as an exposure (Supplementary Table 2) makes it hard to assess whether the lack of significant results using snoring as an exposure was due to a lack of power or due to a lack of a true causal effects. Future efforts could leverage novel statistical genetics methods that use all the GWAS results to test whether the associations observed could be explained by a causality rather than pleiotropy[65].

Finally, we assessed the validity of our GWAS results by using genetic scoring on an independent sample of Australian adults with data on recent snoring. Our successful prediction of snoring using PGS supports the external validity of our genetic association results. Remarkably, we predicted probable OSA using a snoring-derived PGS. Thus, investigating the aetiology of snoring could also help uncover the aetiology and genetic architecture of OSA, a task that has proved to be difficult and challenging[66]. Future efforts could assess the utility of snoring-derived PGS as an addition to the current battery of tests used to more accurately diagnose OSA[67], particularly given the issue of potential OSA underdiagnosis[68,69].

Our results highlight the utility of studying snoring and provide important insights into its aetiology and genetic architecture. However, some limitations must be acknowledged. Analyses used self-reported snoring with the item relying on a partner or close friend complaining about the participant's snoring. Thus, the case definition might be subject to participant-specific recall and subjective biases. Nonetheless, we hypothesize that this limitation might result in the inclusion of some cases as controls (i.e., snoring participants living alone) and therefore bias our results towards the null rather than creating false positives. To avoid confounding due to population stratification, we only included samples of European ancestry in our analyses. This is particularly important, given reports of ethnic differences associated with snoring prevalence[70]. Nonetheless, excluding other populations can limit the generalizability of these results outside the populations studied. As previously discussed, we cannot identify which,

if any, sex-specific genetic observations (e.g., differences in SNP heritability) are due to true genetic effects rather than power differences between the samples. Studying the relationship between snoring and craniofacial phenotypes could provide important insights, given that these traits are likely to share a common aetiology. Nonetheless, there is a limited number of available GWAS summary statistics of craniofacial structure phenotypes. Finally, the fact that PGS for snoring predicted less than 1% of the variance on recent snoring suggest that the GWAS is still underpowered[71]. The heritability for snoring in twin studies is estimated in the range of 18–28%[6], although some of the missing heritability for snoring may come from dominant genetic effects, it is likely that an increased power and studying rare variants[72] yield more powered genetic predictors.

In summary, we provide insights into the aetiology of snoring, its risk factors and genetic underpinnings. Our observational analyses showed a higher prevalence of snoring in males compared with that in females, and effects of age, BMI, SES smoking and alcohol consumption. In addition, tobacco smoking showed a higher effect on snoring prevalence in females compared with that in males, alcohol consumption displayed a higher effect on snoring prevalence in males compared with that in females and SES seemed to be only associated with snoring in males. GWAS identified 127 genome-wide significant associations across 41 genomic risk loci with $h^2_{SNP} = 9.9\%$. We found two loci with differential sex effect sizes, but no evidence for large-scale sex-specific genetic differences. We showed that most of the SNP heritability identified is not simply due to BMI. We also found evidence of a causal relationship from BMI or whole-body fat mass to snoring. Evidence of genetic overlap between snoring and other cardiometabolic, respiratory, neurological and psychiatric traits was found. Finally, we used the GWAS summary statistics to derive individual PGS and predict both recent snoring and probable OSA in an independent sample of Australian adults, thus confirming the relevance of snoring as a sleep-related complex trait. Future studies should aim at leveraging powered GWAS on craniofacial structures, alcohol and tobacco behaviours, to assess whether they are causal of snoring and to assess the amount of shared genetic overlap between OSA and habitual snoring, as the latter may serve to boost the power of obstructive sleep apnoea genetic studies.

## Methods

**Discovery sample and phenotypic information.** Participants included in the present study were of European ancestry from the UK Biobank. Briefly, this resource recruited participants between 2006 and 2010 to assess lifestyle, anthropometric and health-related variables. Participants self-reported on sleep-related traits. Snoring was assessed as a single item (Field-ID: 1210): "Does your partner or a close relative or friend complain about your snoring?" This question could be answered with "Yes", "No", "Don't know", or "Prefer not to answer". We excluded participants whose answers were "Don't know" ($n = 29,309$) or "Prefer not to answer" ($n = 6854$) from our analyses (Supplementary Table 3 shows the total sample size for each GWAS, including sensitivity and sex-stratified analyses). OSA cases were determined on the basis of either ICD-10 diagnosis code or self-report of sleep apnoea diagnosis in the UK Biobank.

**Ethical regulations.** The UK Biobank study was approved by the National Health Service National Research Ethics Service (ref. 11/NW/0382) and all participants provided written informed consent to participate in the UK Biobank study. Information about ethics oversight in the UK Biobank can be found at https://www.ukbiobank.ac.uk/ethics/. Regarding the AGDS, all participants provided informed consent prior to participating in the study. This study and all questionnaires used for AGDS were approved by the QIMR Berghofer Human Research Ethics Committee.

**Data extraction and statistical analyses.** Raw data were extracted from the UK Biobank under Application Number 25331. For a description of the field codes and instances used, refer to Supplementary Table 4. Data were re-coded to remove missing data and uninformative responses (e.g., "I don't know" or "I would rather not answer"). Phenotype-derived estimates such as prevalence and associations

between variables were calculated using *python*. Libraries such as *NumPy* (https://docs.scipy.org/doc/numpy/user/) and *SciPy* (https://docs.scipy.org/doc/) were used for descriptive statistics and *statsmodels* (http://conference.scipy.org/proceedings/scipy2010/pdfs/seabold.pdf) was used to build logistic regression models to assess correlates of snoring and calculate ORs. Snoring prevalence stratified plots (e.g., Fig. 1) were performed using *seaborn* v0.9.0. CIs were calculated using a bootstrapping procedure generating 1000 pseudo replicates of the data.

**Genetic association analyses and QC**. *Discovery GWAS*. Analyses were performed for both the sex-stratified and the full sample using the BOLT-LMM software tool. All GWAS performed were adjusted for age, sex, genotyping array and the first 20 genetic principal components as fixed effects. BOLT-LMM accounts for cryptic relatedness and population stratification. We used a post-GWAS strict QC procedure corresponding to minor allele frequency (MAF ≥ 0.005) and imputation quality (≥0.60).

*Sensitivity analyses*. Given the strong correlation between snoring and BMI, we carried out GWAS for snoring using BMI as a covariate (snoring adj. BMI) with BOLT-LMM software with a QC corresponding to MAF ≥ 0.005 and imputation quality (≥0.60).

**Post-GWAS annotation and functional mapping**. SNP annotation was conducted using the FUMA platform. Risk loci are defined as up to 250 kb based on the most right and left SNPs from each LD block. Gene-based tests were performed using Multi-marker Analysis of GenoMic Annotation (MAGMA) as implemented on the FUMA platform, which provides aggregate association *p*-values based on all variants located within a gene and its regulatory region. We used the GWAS summary statistics to conduct a MAGMA[73] analysis in the FUMA[13] platform (https://fuma.ctglab.nl/). This analysis includes a gene-based test to detect significant SNPs associated with snoring. The prioritized genes based on positional and eQTL mapping were further used to perform gene-set enrichment analysis against the traits available in the GWAS catalogue. Furthermore, we used FUMA to perform tissue-enrichment analysis, based on data from the GTEx project (https://gtexportal.org/home/documentationPage/).

**Genetic correlation analyses**. We performed genetic correlation analyses to estimate genetic correlations between the discovery, sensitivity and sex-stratified snoring GWAS summary statistics using LD score regression (LDSC) as implemented in the Complex Trait Genomics Virtual Lab (CTG-VL, http://genoma.io). Further, to uncover genetically correlated traits with snoring, genetic correlation analyses using LDSC were performed on the platforms CTG-VL and LDHub (http://ldsc.broadinstitute.org/ldhub/), which aggregate summary statistics for GWAS on hundreds of traits.

**Mendelian randomization**. Mendelian randomization (MR) is a method in which genetic variants (e.g., SNPs) are used as instrumental variables to determine causal relationships between traits, environmental exposures, biomarkers or disease outcomes[74]; to satisfy the conditions for MR, it is not required to identify the actual causal variant, because a marker in LD with the causal variant can serve as a proxy instrument[75]. Moreover, to draw conclusions in regard with casual effects, three relevant assumptions must be taken into consideration: (I) genetic variants must be associated with the exposure of interest; (II) genetic variants must not be confounded; (III) genetic variants must be independent of the outcome through other mechanisms[76]. We used GSMR[29], an approach that leverages the usage of multiple independent variables (SNPs) strongly associated with the outcome, to overcome these assumptions, as implemented in the CTG -VL (http://genoma.io). We used GSMR to assess causal relationships between snoring and BMI, whole-body fat mass and pulse rate using our results and existing summary statistics for these traits. To avoid possible confounding from sample overlap, we performed GSMR using the summary statistics derived from sex-stratified GWAS. For example, the female snoring GWAS results were used as exposure, whereas the male pulse rate GWAS results were used for the outcome. Finally, sensitivity two-sample MR analyses (MR-Egger, median-weighted estimator, IVW-MR and weighted mode) were performed using the R library *MRBase*[77] and UK Biobank sex-stratified summary statistics to ensure non-overlapping samples.

**Target sample and polygenic scoring**. To quantify for the cumulative genetic associations for snoring, we calculated PGS using a clumping + thresholding approach. Study description and sample characteristics of the AGDS are available elsewhere[78]. Genotyping was conducted using the Illumina Infinium Global Screening Array platform and genotype imputation using the Haplotype Reference Consortium's reference panel in the Michigan Imputation Server[79] was carried out after performing standard QC procedures. Briefly, for PGS estimation, we excluded indel, strand ambiguous- and low ($R^2$ < 0.6) imputation quality variants. The most significant SNPs were selected using a conservative clumping procedure (PLINK1.9[80]; $p^1 = 1$, $p^2 = 1$, $r^2 = 0.1$, kb = 10,000) to correct for inflation arising from LD. Eight PGS were calculated using different *p*-value thresholds ($p < 5 \times 10^{-8}$, $p < 1 \times 10^{-5}$, $p < 0.001$, $p < 0.01$, $p < 0.05$, $p < 0.1$, $p < 0.5$, $p < 1$) as criteria for

SNP inclusion on the PGS calculation. PGS were calculated by multiplying the effect size of a given SNP by the imputed number of copies (using dosage probabilities) of the effect allele present in an individual. Finally, the SNP dosage effects were summed across all loci per individual. To assess the association between the PGS and snoring and probable OSA, we employed a logistic regression (python *statsmodels*). The target sample for snoring was a subset ($n = 9026$) of the AGDS with data on recent snoring collected through the self-reported item: 'During the last month, on how many nights or days per week have you had or been told you had loud snoring'. The item for probable OSA was: 'During the last month, on how many nights or days per week have you had or been told your breathing stops or you choke or struggle for breath'. For both items, a positive response was considered from one to two times per week up to five to seven times per week and the answer 'Rarely, less than once a week' was excluded. Only a subset ($n \sim 8000$) of highly unrelated individuals (genetic relatedness < 0.05) were included in the analyses.

**Reporting summary**. Further information on research design is available in the Nature Research Reporting Summary linked to this article.

## Data availability
The full GWAS summary statistics for this study will be made available through the NHGRI-EBI GWAS Catalogue (https://www.ebi.ac.uk/gwas/downloads/summary-statistics). Individual level data for UK Biobank participants are available to eligible researchers through the UK Biobank (www.biobank.ac.uk). Individual level AGDS data can be made available to academic collaborators with an appropriate Data Transfer Agreement. Collaboration proposals can be directed to Professor Nick Martin (nick.martin@qimrberghofer.edu.au).

## Code availability
Code used as part of the work presented in this manuscript is available from the authors upon reasonable request.

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

## Acknowledgements

This research was conducted using data from the UK Biobank resource under application number 25331. Data collection for the Australian Genetics of Depression Study was possible, thanks to funding from the Australian National Health & Medical Research Council (NHMRC) to N.G.M. (GNT1086683). A.I.C. is supported by a UQ Research Training Scholarship from The University of Queensland (UQ). M.E.R. thanks the support of the NHMRC and Australian Research Council (ARC), through a NHMRC-ARC Dementia Research Development Fellowship (GNT1102821). G.C.-P. is funded by an ARC Discovery Early Career Researcher Award (DE180100976). We thank our colleague Jackson Thorp for proof-reading our manuscript and useful feedback and discussion.

## Author contributions

M.E.R. and G.C.-P. conceived and directed the study. A.I.C. and L.M.G.-M. performed most of the statistical and bioinformatics analyses, with support from G.C.-P. and M.E.R. E.M.B. and N.G.M. collected and contributed data from the Australian sample. A.I.C., L.M.G.-M., M.E.R. and G.C.-P. wrote the paper with feedback from all co-authors.

## Competing interests

The authors declare no competing interests.
