## [Peer Review File · Nature Communications]

Reviewers' Comments:

Reviewer #1:

Remarks to the Author:

This is an interesting and well-written manuscript addressing the genetic basis of snoring. The authors used data from British individuals as part of the UK Biobank and validated some of the findings in an independent Australian cohorts. I have a couple of suggestions to further improve the paper:

1) It is intriguing that additional adjustment for body mass index did not do much to the main findings of the study. As BMI is only an overall measure of adiposity, I would also consider additional adjustment for measures indicating increased central adiposity such as waist circumference or waist-to-hip ratio adjusted for body mass index (which are both available in the UK Biobank). This might give different results and will provide additional information on the role of adiposity in these findings (given that some of the loci were mapped to BMI-associated genes, such as FTO).

2) The authors performed bidirectional Mendelian randomization analyses using summary-based statistics. The authors are not really clear about the used methodology, but highlighted that there is still potential bias due to pleiotropic effects. This is certainly an issue, which could already be addressed in the present paper. There are multiple statistical techniques described in the literature to provide more unbiased estimates in two-sample Mendelian Randomization analyses. These techniques include MR-Egger and Median-weighted estimator (among some other techniques which were more recently described). The authors should perform these sensitivity analyses in their revised manuscript. There are multiple papers in the literature that explain how to perform these kind of analyses, and most of them have been implemented in the MRBase package.

3) The analyses presented in Figure 1 are interesting, but these are unadjusted for confounding factors. Furthermore, I am not sure what the specific added value is of these analyses with respect to the rest of the paper. A lot of emphasis is given on alcohol, but these findings do not add much to the rest of the paper.

4) Figure 2. The present figure is mainly focused on p-values whereas it might be interesting to provide more insights in the differences in beta estimates in the two GWAS (adjusted and unadjusted for body mass index). Especially the ones that show clear reduction in effect size are not better presented.

Reviewer #2:

Remarks to the Author:

In this paper the authors Campos et al. perform a large GWAS in the frequency of snoring in the UKBB sample. They take into account proper covariates and major environmental risk factors, examine sex-specific associations and examine PRS correlations in an independent sample. The findings are interesting in light of the genetic determinants of snoring but especially since snoring is a strong proxy for sleep apnea. I have only a few comments to address for this paper.

Based on the discoveries in this paper, snoring presents as 1) strongly predictive for OSA but also 2) as a highly pleiotropic trait. Indeed the genetic correlations anchor snoring with multitude of health outcomes as well as educational attainment, alcohol use, smoking and BMI (a major risk factor for OSA). While BMI is correctly addressed in the present study, the authors should examine snoring in different socioeconomic groups at the epidemiological level. In addition, the authors should examine how socioeconomic status shapes the genetic associations, if there is evidence from MR from lower socioeconomic status leading to snoring, or evidence for gender specific effects.

The association with sleep apnea is exciting. As it is likely under-reported in UKBB it would be

interesting to see genetic correlation with previously reported proxies for OSA with GWAS available (<http://sleepdisordergenetics.org/>): average SpO₂, minimum SpO₂ and Per90. Similarly, do any of the findings replicate cross-trait in this independent data set?

Definition of snoring is weird as while correct, everyone's muscles relax during sleep and yet not everyone snores. Reformat.

Line 253 in discussion: "Our results highlight the relevance of studying snoring as an independent trait." To me the excitement in the paper is how well snoring captures OSA in general population and not necessarily snoring on its own. Maybe worth summarizing what are the reasons to study snoring, if it is purely clinical correlations with OSA, or whether there are other aspects to consider.

The similarity between men and women is interesting. Maybe snoring is different from OSA and reflects larger shared pathology between sexes. Please also indicate was SNP based heritability similar in males vs. females, possibly environmental factors play larger role in males than in females?

Reviewer #3:

Remarks to the Author:

In this work, the authors take advantage of large GWA samples to explore the genetics of snoring. Snoring is strongly associated with sleep apnea. Significant loci are uncovered and mendelian randomization is conducted to look at associations with BMI. A causal relationship with an increased pulse was also noted using MR.

One weakness is how BMI (and to a lesser extent age) is dealt with. It is unclear how this was done (methods are not well described) but it is not sufficient to include BMI as a covariate to explore the effects of BMI on associated variants.

It is well established that controlling for the effect of BMI does not control fully for weight effects on sleep-disordered breathing. The model needs to be adjusted for more complex effects such as age, age², sex, age × sex, BMI, and BMI² to address known demographic factors and potential non-linear effects of age and BMI on snoring. Also, I don't understand why the authors did not add other models that also include whole-body fat.

In this direction, Figure 3 should absolutely also present genetic correlations corrected in more definitive ways by age, sex, and BMI as discussed above.

Similarly, considering the large number of subjects, I would suggest exploring the results of a sub-analysis in subjects with low BMI or underweight, for example with BMI <18.5 and BMI <20. Indeed, at low BMI effects, it is likely that craniofacial genetic effects may start to appear, and it is surprising this is not explored or discussed. Maybe a different genetic picture could emerge.

Then, the manuscript is thin with regard to exploring or discussing causal relationships of snoring as a proxy for sleep apnea in causing high blood pressure or in mediating increased cardiovascular events (heart attacks, stroke, CAD etc), independent of BMI controlled as discussed above. A causal relationship with pulse is found, what about BP, which is highly correlated per Figure 3? A negative result using MR would also be important, although of course snoring is not sleep apnea. In this regard, it is unclear if the results are not presented in Figure 5, suppl figure 4, suppl table 1, because they are not significant or if it was never tested. If it has been done and is not significant, it should still be presented.

Looking at the genetic overlap with the genetics of craniofacial structure, or allergy (nose permeability is a risk factor for snoring) is also lacking.

Otherwise, I found the manuscript to be well written and to the point; it would just benefit from additional analyses that are pertinent to sleep disordered breathing researchers.

Then, it is also a very thin with regard to exploring casual relationships of snoring as a proxy for sleep apnea in causing high blood pressure or in mediating increased cardiovascular events (heart attacks etc) independent of BMI controlled as discussed above. A causal relationship with pulse is found, what about BP, which is highly correlated with snoring per Figure 3? A negative result using MR would also be important, although of course snoring is not sleep apnea.

Looking at the genetic overlap with the genetics of craniofacial structure, or of allergy (nose permeability is a risk factor for snoring) is also lacking.

Otherwise, I found the manuscript to be well written and to the point; it would just benefit from additional analysis that are pertinent to sleep disordered breathing researchers.

RESPONSE TO REVIEWER'S COMMENTS

Reviewer #1 (Remarks to the Author):

This is an interesting and well-written manuscript addressing the genetic basis of snoring. The authors used data from British individuals as part of the UK Biobank and validated some of the findings in an independent Australian cohort. I have a couple of suggestions to further improve the paper:

1) It is intriguing that additional adjustment for body mass index did not do much to the main findings of the study. As BMI is only an overall measure of adiposity, I would also consider additional adjustment for measures indicating increased central adiposity such as waist circumference or waist-to-hip ratio adjusted for body mass index (which as both available in the UK Biobank). This might give different results and will provide additional information on the role of adiposity in these findings (given that some of the loci were mapped to BMI-associated genes, such as FTO).

We thank the reviewer for raising this important concern. The extent to which our approach was effectively correcting for adiposity in general was consistently raised by the reviewers as an issue. To address this issue, we have now performed additional analyses and revised the paper accordingly.

Our original approach was indeed effective at removing genetic effects associated with BMI, as evidenced by the fact that BMI and other adiposity related traits were no longer genetically correlated with the “snoring-adjustedforBMI” summary statistics. The main text now reads “*BMI and adiposity related phenotypes did not show a significant genetic correlation with snoring after adjusting for BMI*”. Figure 3 was also modified to reflect this.

We performed an additional sensitivity GWAS adjusting for the covariates suggested by the reviewers: BMI, BMI², and whole body-fat mass (WBFM), as well as age² and age*sex. Simultaneously controlling for both BMI and WBFM effectively controls for the effect of one covariate while keeping the other constant (i.e. adjusting for each other). The results of this stricter analysis showed an LDSC genetic correlation of 0.9998 (S.E.=0.0002) with the original BMI-adjusted snoring GWAS, suggesting that any genetic effects observed were not driven by these other covariates. The revised section of the manuscript now reads: “We performed two follow-up sensitivity GWAS for snoring. The first one including BMI as a covariate, and the second one including BMI, BMI², age*sex, age² and whole-body fat mass to explore the effects of BMI, adiposity and non-linear effects on associated variants. Both sensitivity analyses showed very similar results, with a genetic correlation of 0.9998 non-significantly different from 1 (S.E.=0.0002).”

2) The authors performed bidirectional mendelian randomization analyses using summary-based statistics. The authors are not really clear about the used methodology, but highlighted that there is still potential bias due to pleiotropic effects. This is certainly an issue, which could already be addressed in the present paper. There are multiple statistical techniques described in the literature to provide more unbiased estimates in two-sample Mendelian Randomization analyses. These techniques include MR-Egger and Median-weighted estimator (among some other techniques which were more recently described). The authors should perform these sensitivity analyses in their revised manuscript. There are multiple papers in the literature that explain how to perform these kinds of analyses, and most of them have been implemented in the MRBase package.

We revised Supplementary Table 2 to include results from not only GSMR (our initial approach), but also from 4 other methods, including those suggested by the reviewer. While Median Weighted Estimator, GSMR and Inverse Variance Weighted MR supported the existence of a causal effect of BMI on snoring, the MR-Egger estimate did not reach statistical significance (perhaps due to MR-Egger being less powered in comparison to other methods, see Schmidt and Dudbridge 2018). Nonetheless, the MR-Egger intercept was not significantly different from zero, which suggests that the IVW estimate can be relied upon as it is unlikely to be biased by pleiotropy (Burgess & Thompson 2017). We have now described these new results in the relevant sections of the manuscript, and revised Figure 5 to reflect the results from the 5 different MR methods used to assess the causal relationship between BMI and snoring.

3) The analyses presented in Figure 1 are interesting, but these are unadjusted for confounding factors. Furthermore, I am not sure what the specific added value is of these analyses with respect to the rest of the paper. A lot of emphasis is given on alcohol, but these findings do not add much to the rest of the paper.

We thank the reviewer for this comment. The results presented in Figure 1a were derived using a multivariate logistic regression analysis which simultaneously estimated the effects of all covariates in the model (age, BMI, Townsend deprivation index, household income, tobacco usage and alcohol frequency). The motivation to present these results stem from the fact that the phenotypic richness of the UK Biobank enables us to study the effect sizes of known or suspected risk factors for snoring in a sample of unprecedented scale. Our detailed analyses showed that some of the known environmental risk factors actually display sex differences (see for example the new analyses on socio-economic status). This observation could partly explain the observed gender differences on snoring prevalence. Finally, the phenotypic associations relating snoring to alcohol intake and smoking behavior were

also recapitulated as genetic correlations. We mention and discuss the above points in detail in the revised version of the discussion.

4) Figure 2. The present figure is mainly focused on p-values whereas it might be interesting to provide more insights in the differences in beta estimates in the two GWAS (adjusted and unadjusted for body mass index). Especially the ones that show clear reduction in effect size are than better presented.

We have now updated Figure 2 to compare the effect sizes between both GWAS. Consistent with the reviewer's first suggestion, the effect size of the signal mapping to the FTO locus disappeared after adjusting for BMI. This was the only statistically significant (after multiple testing correction) change observed. We have further modified the text describing and discussing these observations. We would like to thank the reviewer as his comments and suggestions have helped us produce a higher quality, clearer manuscript. See below the plot in Figure 2b:

Reviewer #2 (Remarks to the Author):

In this paper the authors Campos et al. perform a large GWAS in the frequency of snoring in the UKBB sample. They take into account proper covariates and major environmental risk factors, examine sex-specific associations and examine PRS correlations in an independent sample. The findings are interesting in light of the genetic determinants of snoring but especially since snoring is a strong proxy for sleep apnea. I have only a few comments to address for this paper.

We thank the reviewer for his/her time assessing our manuscript and appreciate his/her constructive feedback.

Based on the discoveries in this paper, snoring presents as 1) strongly predictive for OSA but also 2) as a highly pleiotropic trait. Indeed, the genetic correlations anchor snoring with multitude of health outcomes as well as educational attainment, alcohol use, smoking and BMI (a major risk factor for OSA). While BMI is correctly addressed in the present study, the authors should examine snoring in different socioeconomic groups at the epidemiological level. In addition, the authors should examine how socioeconomic status shapes the genetic associations, if there is evidence from MR from lower socioeconomic status leading to snoring or evidence for gender specific effects

We thank the reviewer for this interesting suggestion. In the revised manuscript, we include both *Townsend deprivation index* and *Average annual household income* as proxy-variables for socioeconomic status (SES). The results are shown in Figure 1 and discussed throughout the revised manuscript. Interestingly, SES effect was only significant for males and in the direction the reviewer suspected (lower SES associated with higher snoring prevalence) but not in females. Given that our model simultaneously takes also into account the effects of BMI, alcohol and tobacco consumption frequency, we speculate that the snoring-SES association might potentially be driven by differences in workplace environment or other lifestyle variables that differ between low-SES males and females. The Supplementary Material describes a small but significant genetic correlation between snoring and these measures. Unfortunately, as it was also the case for alcohol and smoking, the SES measures are relatively underpowered to perform Mendelian randomization analyses (e.g. a GSMR with Townsend deprivation index as exposure would only have four SNPs available; see figure below) to assess a possible causal relationship of SES on snoring.

The association with sleep apnea is exciting. As it is likely under-reported in UKBB it would be interesting to see genetic correlation with previously reported proxies for OSA with GWAS available (<http://sleepdisordergenetics.org/>): average SpO2, minimum SpO2 and Per90. Similarly, do any of the findings replicate cross-trait in this independent data set?

We calculated genetic correlation between snoring and the previously reported proxies for OSA, as suggested by the reviewer. Interestingly, snoring was genetically correlated with minimum SpO2 and Per90, but not with average SpO2. These correlations remained significant after adjusting for BMI. We have now added a sentence about these observations in the results section of the paper and modified Figure 3.

Definition of snoring is weird as while correct, everyone's muscles relax during sleep and yet not everyone snores. Reformat.

We have modified the definition to avoid ambiguity. The main text now reads: "Snoring is the vibration of upper airway structures that occurs during sleep and creates noise as the air passes in and out while breathing"

Line 253 in discussion: "Our results highlight the relevance of studying snoring as an independent trait." To me the excitement in the paper is how well snoring captures OSA in general population and not necessarily snoring on its own. Maybe worth summarizing what are the reasons to study snoring, if it is purely clinical correlations with OSA, or whether there are other aspects to consider.

We agree with the reviewer in that snoring represents a well-powered proxy-phenotype for studying sleep apnoea. However, since snoring is not the only symptom of obstructive sleep apnoea, we tried to discuss throughout the manuscript how studying individual symptoms can help unveil the heterogeneity and better understand the biology of a disorder. We also decided to reword that sentence and remove "as an independent trait" to avoid confusion and better reflect the importance of snoring as a cardinal symptom of sleep apnea.

The similarity between men and women is interesting. Maybe snoring is different from OSA and reflects larger shared pathology between sexes. Please also indicate was SNP based heritability similar in males vs. females, possibly environmental factors play larger role in males than in females?

The reviewer raises an interesting point for discussion. Consistent with the reviewer's hypothesis, we identified a higher SNP heritability for females (12.42%) than for males (8.77%). This is shown in Table 3. We did not identify any large-scale sex-specific genetic effects, as the overall genetic correlation between the sex-stratified GWAS was high ($r_G=0.914$, S.E.=0.033, p-value= 1.91×10^{-10}).

160) and effect sizes and directions for top hits were highly consistent in both the male and female samples (Supplementary File 1 and Supplementary Figure 5). Nonetheless, we identified two signals with a statistically significant sex specific effect. We have revised Supplementary Figure 5 to more accurately show this. Also, we added a mention of the differences in SNP heritability and the loci with different effect sizes on males and females in the Results and Discussion.

Reviewer #3 (Remarks to the Author):

In this work, the authors take advantage of large GWA samples to explore the genetics of snoring. Snoring is strongly associated with sleep apnea. Significant loci are uncovered and mendelian randomization is conducted to look at associations with BMI. A causal relationship with an increased pulse was also noted using MR. One weakness is how BMI (and to a lesser extent age) is dealt with. It is unclear how this was done (methods are not well described) but it is not sufficient to include BMI as a covariate to explore the effects of BMI on associated variants. It is well established that controlling for the effect of BMI does not control fully for weight effects on sleep-disordered breathing. The model needs to be adjusted for more complex effects such as age, age², sex, age × sex, BMI, and BMI² to address known demographic factors and potential non-linear effects of age and BMI on snoring. Also, I don't understand why the authors did not add other models that also include whole-body fat.

We thank the reviewer for raising this issue. All GWAS reported in the manuscript included the following standard covariates: age, sex, genotyping array, and the first 20 genetic ancestry principal components (PCs) to correct for population stratification. We have revised the Methods section to explicitly mention this. Also, as we have argued above (in response to point 1 of Reviewer 1), the fact that the genetic correlation of snoring with BMI and other adiposity traits disappears in the GWAS of snoring adjusted for BMI, is strong evidence that including BMI as a covariate in the model successfully accounts for BMI effects. This is reflected in the revised Figure 3 and discussed in the main text.

Furthermore, we also performed a stricter sensitivity analysis including age, sex, age*sex, age², genotyping array, 20PCs, BMI, BMI² and Whole-Body Fat Mass as covariates. Our results showed no changes, as evidenced by a very high genetic correlation (non-significantly different from 1 $r_g=0.9998$, S.E. =0.0002) with the previous (BMI + basic covariates adjusted) GWAS. We have included a description of this analysis in the results.

In this direction, Figure 3 should absolutely also present genetic correlations corrected in more definitive ways by age, sex, and BMI as discussed above.

We thank the reviewer for raising this issue. We understand the importance of assessing genetic correlations after adjusting for BMI (and other adiposity measures). The revised manuscript includes a revised Figure 3 as suggested by the reviewer. As expected, some traits (particularly those related to adiposity) showed great changes in their genetic correlations with snoring after adjusting for BMI. We would like to note though, that correcting for heritable covariates in a GWAS is prone to introducing collider bias (see Aschard *et al.* 2015 and Zhihong Zhu *et al.* 2018), and not only reduces power, but makes the interpretation of the results very difficult, particularly (but not restricted to) when new genetic correlations emerge after the correction. In fact, the proper way to perform such an analysis would require adjusting for BMI in all other traits for which we wish to perform genetic correlations or MR studies (see for example 10.1093/schbul/sby162). Given that adjustment for BMI was a sensitivity analysis of the original GWAS, we consider that such task is well beyond the scope of this manuscript, but certainly an interesting approach for a subsequent study.

Similarly, considering the large number of subjects, I would suggest exploring the results of a sub-analysis in subjects with low BMI or underweight, for example with BMI <18.5 and BMI <20. Indeed, at low BMI

effects, it is likely that craniofacial genetic effects may start to appear, and it is surprising this is not explored or discussed. Maybe a different genetic picture could emerge.

The reviewer raises a very interesting possible sub-analysis. We looked at the UK Biobank data and unfortunately there were only ~10,000 individuals with a BMI \leq 20 of which only 1,637 report snoring. Therefore, the analyses would clearly be underpowered. Nonetheless, we have provided in this revised manuscript further evidence that our sensitivity analysis effectively corrects for the effect of BMI, and in fact most of the signals identified are not simply explained by BMI.

Then, the manuscript is thin with regard to exploring or discussing causal relationships of snoring as a proxy for sleep apnea in causing high blood pressure or in mediating increased cardiovascular events (heart attacks, stroke, CAD etc), independent of BMI controlled as discussed above. A causal relationship with pulse is found, what about BP, which is highly correlated per Figure 3? A negative result using MR would also be important, although of course snoring is not sleep apnea. In this regard, it is unclear if the results are not presented in Figure 5, suppl figure 4, suppl table 1, because they are not significant or if it was never tested. If it has been done and is not significant, it should still be presented.

This is certainly an interesting point. We performed additional GSMR analysis between snoring and (i) heart attack, (ii) diastolic blood pressure and (iii) a major coronary heart disease event. Interestingly there was evidence for snoring having a causal effect on diastolic blood pressure, and a bidirectional causal relationship between snoring and heart attack, with a stronger evidence (but also more power) for snoring causing heart attack. On the contrary, we found no evidence for a causal effect on major coronary heart disease. We have now included these results, along with a range of MR methods as sensitivity analyses, as part of supplementary figure 4, supplementary Table 2 and discussed them within the main text. Nonetheless, the results were not significant after accounting for sample overlap and multiple testing.

Looking at the genetic overlap with the genetics of craniofacial structure, or allergy (nose permeability is a risk factor for snoring) is also lacking.

We thank the reviewer for raising this point. None of the tested measures for allergy (i.e. allergic rhinitis nor asthma) were genetically correlated with snoring. This could imply that the association between allergic rhinitis and snoring is not driven through genetic mechanisms. Given that this observation warrants a closer inspection, we have added a brief sentence about this in the discussion section. Regarding craniofacial structure traits, unfortunately we have not been able to access GWAS summary statistics for measures of craniofacial structure. We agree that this is an important analysis and have added a brief mention of it as a perspective for future studies.

Reviewers' Comments:

Reviewer #1:

Remarks to the Author:

The authors revised the manuscript according to my suggestions. This is a very nice manuscript that will add much to the existing literature.

Reviewer #2:

Remarks to the Author:

In this paper the authors have now formatted the paper and included additional analyses such as those for socioeconomic status and those for previous apnea measures. I do not have any further comments.

Reviewer #3:

Remarks to the Author:

Having added the study of other cardiovascular consequences is nice. I'm also generally pleased by the revision but still somewhat surprised that the results changed so little except for FTO when controlled better for BMI and in a more complex way with age.

I still wish the authors would have done more. The relationships between snoring, sleep apnea, craniofacial structure, age and BMI are complex. Tissue flaccidity increases with high-end age and is likely to produce a different type of problem. Increased weight probably interacts with the various genetic factors in slightly different ways. The fact that there are so few people snoring at BMI<20 (underweight) is revealing. It would have been worth looking at snoring in the BMI< 23 subgroup (normal and underweight), and when I suggested to look at BMI<20, it was not to be meant to be an exact value, more a direction of research to make the author do more, and maybe reveal more the genes they suggest are correlated with musculoskeletal traits.

Another minor comment, there is nothing surprising with snoring correlating better with min O2 saturation and per 90 versus mean Sao2. Mean SaO2 mostly reflects other problems such hypoventilation, not sleep apnea (although low mean saO2 occurs in obesity hypoventilation syndrome that often also has sleep apnea). This is a misunderstanding of basic physiology.

The sentence " Notably, the OSA proxy traits (minimum SpO2, average SpO2 and Perc90) showed small increases in their genetic correlation with snoring after adjusting for BMI (Figure 3 and Supplementary Data 3)."

may thus be more accurate as

Notably, measures of nocturnal hypoxia known to correlates with sleep apnea severity such as minimum SpO2 and perc 90, showed small increases in their genetic correlation with snoring after adjusting for BMI (Figure 3 and Supplementary Data 3).".

Similarly,

We also analysed the genetic correlation between snoring and three measures of overnight oxyhemoglobin saturation (average SpO2, minimum SpO2 saturation and percent of sleep with oxyhemoglobin saturation under 90% Perc90), which are known proxies for sleep disordered breathing. Minimum SpO2 and Perc90, but not average SpO2, showed moderate significant genetic 90 correlations with snoring (Figure 3).

is also incorrect and could be:

We also analysed the genetic correlation between snoring and three measures of overnight oxyhemoglobin saturation: mean SpO₂, minimum SpO₂ and percent of sleep with oxyhemoglobin saturation under 90% Perc90. Minimum SpO₂ and Perc90, which are known proxies for sleep disordered breathing, but not average SpO₂, which reflect decreased ventilation not generally related to sleep apnea, showed moderate significant genetic correlations with snoring (Figure 3).

RESPONSE TO REVIEWERS' COMMENTS:

Reviewer #1 (Remarks to the Author):

The authors revised the manuscript according to my suggestions. This is a very nice manuscript that will add much to the existing literature.

Response: We thank the reviewer for the useful comments and taking his/her time again to assess our manuscript.

Reviewer #2 (Remarks to the Author):

In this paper the authors have now formatted the paper and included additional analyses such as those for socioeconomic status and those for previous apnea measures. I do not have any further comments.

Response: We thank the reviewer for the useful comments and taking his/her time again to assess our manuscript.

Reviewer #3 (Remarks to the Author):

Having added the study of other cardiovascular consequences is nice. I'm also generally pleased by the revision but still somewhat surprised that the results changed so little except for FTO when controlled better for BMI and in a more complex way with age.

Response: We thank the reviewer for assessing our manuscript. While a greater genetic overlap between BMI and snoring could be expected given their strong dependency, family studies of snoring (e.g. Carmelli, 2001; PID: 11587976) have provided evidence for a largely independent genetic factors influencing snoring and obesity in older males. Our results extend these observations to a younger age, both sexes and to the general relationship between BMI (not only obesity) and snoring.

I still wish the authors would have done more. The relationships between snoring, sleep apnea, craniofacial structure, age and BMI are complex. Tissue flaccidity increases with high-end age and is likely to produce a different type of problem. Increased weight probably interacts with the various genetic factors in slightly different ways. The fact that there are so few people snoring at BMI<20 (underweight) is revealing. It would have been worth looking at snoring in the BMI< 23 subgroup (normal and underweight), and when I suggested to look at BMI<20, it was not to be meant to be an exact value, more a direction of research to make the author do more, and maybe reveal more the genes they suggest are correlated with musculoskeletal traits.

Response: As stated in the previous response, this is an interesting proposition and we regret that the sample size is likely to limit our findings. For example, the sex stratified GWAS (which reduced sample sizes roughly by half) were significantly underpowered when compared to the main analysis.

Another minor comment, there is nothing surprising with snoring correlating better with min O2 saturation and per 90 versus mean Sao2. Mean SaO2 mostly reflects other problems such hypoventilation, not sleep apnea (although low mean saO2 occurs in obesity hypoventilation syndrome that often also has sleep apnea). This is a misunderstanding of basic physiology.

The sentence "Notably, the OSA proxy traits (minimum SpO2, average SpO2 and Perc90) showed small increases in their genetic correlation with snoring after adjusting for BMI (Figure 3 and Supplementary Data 3)." may thus be more accurate as "Notably, measures of nocturnal hypoxia known to correlates with sleep apnea severity such as minimum SpO2 and perc 90, showed small increases in their genetic correlation with snoring after adjusting for BMI (Figure 3 and Supplementary Data 3)."

Response: This is a great point. We have now modified the sentence to read as suggested: “Notably, measures of nocturnal hypoxia such as minimum SpO₂, average SpO₂ and Perc90, showed small increases in their genetic correlation with snoring after adjusting for BMI (Figure 3 and Supplementary Data 3).”

Similarly,

We also analysed the genetic correlation between snoring and three measures of overnight oxyhemoglobin saturation (average SpO₂, minimum SpO₂ saturation and percent of sleep with oxyhemoglobin saturation under 90% Perc90), which are known proxies for sleep disordered breathing. Minimum SpO₂ and Perc90, but not average SpO₂, showed moderate significant genetic 90 correlations with snoring (Figure 3). is also incorrect and could be: *We also analysed the genetic correlation between snoring and three measures of overnight oxyhemoglobin saturation: mean SpO₂, minimum SpO₂ and percent of sleep with oxyhemoglobin saturation under 90% Perc90. Minimum SpO₂ and Perc90, which are known proxies for sleep disordered breathing, but not average SpO₂, which reflect decreased ventilation not generally related to sleep apnea, showed moderate significant genetic 90 correlations with snoring (Figure 3).*

Response: We thank the reviewer for this suggestion, we have now modified the sentence to acknowledge that average SpO₂ reflects changes in ventilation that are not necessarily related to sleep disordered breathing. The sentence now reads: “*Minimum SpO₂ and Perc90, which are known proxies for sleep disordered breathing, but not average SpO₂ (which reflects changes in ventilation not necessarily related to sleep apnoea), showed moderate significant genetic correlations with snoring (Figure 3).*”